



**Influence of slope aspect on the microbial properties of rhizospheric**
**and non-rhizospheric soil on the Loess Plateau, China**
Ze Min Ai [1,2,3], Jiao Yang Zhang [1,2,3], Hong Fei Liu [4], Sha Xue [1,2]*, Guo Bin Liu [1,2]

[1] State Key Laboratory of Soil Erosion and Dryland Farming on the Loess Plateau, Institute of Soil and
Water Conservation, Northwest A&F University, Yangling, 712100, People's Republic of China
[2] Institute of Soil and Water Conservation, Chinese Academy of Sciences & Ministry of Water
Resources, Yangling, Shaanxi 712100, People's Republic of China
[3] University of Chinese Academy of Sciences, Beijing 100101, People's Republic of China
[4] College of Forestry, Northwest A&F University, Yangling, Shaanxi 712100, People's Republic of
China

* Corresponding author.
E-mail address: xuesha100@163.com.
*Corresponding address: Xinong Rd. 26, Institute of Soil and Water Conservation, Northwest A&F
University, Yangling, Shaanxi 712100, People's Republic of China.





**Abstract.** Slope aspect is an important topographic factor, but its effect on the microbial properties of
grassland rhizospheric soil (RS) and non-rhizospheric soil (NRS) remain unclear. A field experiment
was conducted at the Ansai Research Station on the Loess Plateau in China to test the influence of
slope aspects (south-facing, north-facing, and northeast-facing slopes, all with *Artemisia sacrorum* as
the dominant species) on RS and NRS microbial biomass carbon (MBC) and phospholipid fatty acid
(PLFA) contents, and the rhizospheric effect (RE) of various microbial indices. MBC content differed
significantly among the slope aspects in RS but not in NRS, and RE for MBC content in the
south-facing slope was larger than that in the north-facing slope. RS total, bacterial, and gram-positive
bacterial PLFA contents in the south-facing slope were significantly lower than those in the north- and
northeast-facing slopes, and RS gram-negative bacterial (G⁻) and actinomycete PLFA contents in the
south-facing slope were significantly lower than those in the north-facing slope. Differently, NRS total,
bacterial, and G⁻ PLFA contents in the north-facing slope were significantly higher than those in the
south- and northeast-facing slopes, and NRS fungal and actinomycete PLFA contents in the north- and
south-facing slopes were significantly higher than those in the northeast-facing slope. RE for all PLFA
contents except fungal in the northeast-facing slope were higher than those in the south-facing slope.
Slope aspect significantly but differentially affected the microbial properties in RS and NRS, and the
variable influence was due an evident RE for most microbial properties.
**Keywords:** topographic factor, rhizospheric effect, phospholipid fatty acid, fungi, bacteria,
actinomycete

## 1    Introduction

As an important topographic factor, slope aspect can affect the amount of solar radiation received and
the angle between the ground and wind direction, which is defined as the orientation faced by a slope
(Selvakumar et al., 2009). Solar radiation influences ecologically critical factors of local microclimates
and determines soil temperature, evaporation capacity, and soil-moisture content (Carletti et al.,
2009;Bennie et al., 2008). Slope aspect can substantially affect soil-moisture content, water budget, and
soil temperatures (Sidari et al., 2008;Wang et al., 2011;Carletti et al., 2009;Sariyildiz et al., 2005).
South-facing slopes in the Northern Hemisphere, which receives the more solar radiation than
north-facing slopes, are typically hot, dry, and subject to rapid changes in seasonal and diurnal
microclimates. North-facing slopes are the opposite, which receive the least insolation, are cool, moist,
and subject to slow changes in seasonal and daily microclimates (Sariyildiz et al., 2005). The effect of
slope aspect on basic soil properties (pH, bulk density, and texture), nutrient (carbon, nitrogen, and
phosphorus) contents, microbial biomass, and enzymatic activities have been studied (Ai et al.,
2017a;Ascher et al., 2012;Gilliam et al., 2014;Huang et al., 2015;Sidari et al., 2008;Qin et al., 2016).
North-facing slopes have more microbial biomass carbon (MBC), bacteria, and actinomycetes than
south-facing slopes (Ascher et al., 2012;Huang et al., 2015); in contrast, other studies have found that
MBC and total and fungal phospholipid fatty acid (PLFA) contents were significantly higher in
south-facing than north-facing slopes (Huang et al., 2015;Sidari et al., 2008;Gilliam et al., 2014).
Gilliam et al. (2014) found that bacterial biomass did not vary with slope aspect. The different results
of these studies may have been due to the differences in plant species (trees vs shrubs), soil properties,
climatic conditions, and research methods. Previous studies mainly focused on trees and shrubs, but the
influence of slope aspect on grassland soil microorganisms is still unclear, even though the grassland
ecosystem is an important component of terrestrial ecosystems.



The rhizosphere is commonly defined as the narrow zone of soil adjacent to and influenced by plant roots (Chen et al., 2002). The rhizosphere contains root exudates, i.e. leaked and secreted chemicals, sloughed root cells, and plant debris (Warembourg et al., 2003). Microbial activity is therefore high in rhizospheric soil (RS) and clearly distinct from the activity in non-rhizospheric soil (NRS) due to differences in nutrient availability, pH, and redox potential (Hinsinger et al., 2009). Microbial content is higher in RS than NRS (Buyer et al., 2002;Marschner et al., 2002), which is known as the rhizospheric effect (RE). The effect of slope aspect on RS and NRS microbial biomass and composition has not been extensively studied. Knowledge of the influence of slope aspect on the differences between RS and NRS microbial communities could provide new insights into topographical influences of RE on local micro-ecosystemic environments.

Soil microbial communities play important roles in soil quality and ecosystemic processes, including nutrient cycling, decomposition of organic matter, bioremediation of structural formation, and even plant interactions (Harris, 2009). These communities are closely associated with their surroundings, rapidly responding to changes and environmental stresses. Soil microbes are thus commonly used as sensitive indicators of change to soil quality under environmental stresses. Soil respiration is widely used for measuring microbial activity (e.g. basal respiration) or determining the potential microbial activity in soil (e.g. substrate-induced respiration) (Nannipieri et al., 1990;Wardle, 1995). Various microbial PLFAs represent the different nutritional requirements of the microbial groups. Bacteria and fungi form most of the microbial biomass and represent the main drivers of organic-matter turnover (Bååth and Anderson, 2003). Moreover, different kinds of bacteria produce different PLFAs: Gram-negative ($G^-$) and Gram-positive ($G^+$) bacterial PLFA contents are usually considered indicators of chemolithotrophic and heterotrophic bacterial communities, respectively. $G^-$ bacteria are mainly associated with roots and thus decompose low-molecular-weight organic molecules (Griffiths et al., 1999), whereas $G^+$ bacteria decompose more complex materials, such as organic matter and litter (Kramer and Gleixner, 2006). These microbial indices are all sensitive bio-indicators that can be used to estimate soil quality and the effect of slope aspect on RS and NRS microbial communities. Soil ecologists have long been interested in the response of microbial communities to environmental factors for understanding the underlying mechanisms determining the content and composition of microbial biomass. Microbial communities have a close relationship with pH, carbon (organic and water-soluble organic carbon), nitrogen (total nitrogen, ammonium and nitrate nitrogen, and water-soluble ammonium and nitrate nitrogen), and phosphorus (total and available phosphorus) (Bardelli et al., 2017;Huang et al., 2014;Nilsson et al., 2005;Ma et al., 2015). The effect of slope aspect on the main soil nutrient factors that affect RS and NRS microbial communities, however, remains unclear.

The Chinese government introduced the Grain for Green Project in the 1990s to control soil erosion and improve the ecological environment of the Loess Plateau by converting large areas of sloping cropland to forest and grassland. *Artemisia sacrorum*, a perennial herb with multiple branches, well-developed root suckers, and high seed production and fertility, is widely distributed on the plateau (Wang and Liu, 2002), especially in the converted grassland. We selected *A. sacrorum* as a typical grassland plant of this region to study the effect of slope aspect on the MBC, total, fungal, bacterial, and actinomycete PLFA contents in RS and NRS and the differences of their REs. We also identified the main RS and NRS environmental factors affecting microbial content and composition. We tested three slope aspects (south-facing, north-facing, and northeast-facing slopes) with the same rehabilitation age on the Loess Plateau in China. We tested the following hypotheses: (1) slope aspect would significantly but differentially affect the MBC, total, fungal, bacterial, and actinomycete PLFA





contents and their REs; and (2) soil carbon (C) and nitrogen (N) would have the larger effect on the RS
and NRS microbial communities, and different C and N compounds would have different effects.
**2 Materials and methods**
**2.1 Study site**
A field experiment was conducted at the Ansai Research Station (ARS) of the Chinese Academy of
Sciences (36°51′30″N, 109°19′23″E; 1068–1309 m a.s.l.), northern Loess Plateau, China. The mean
annual temperature in the study area is 8.8 ℃, and the mean annual precipitation is approximately 505
mm, with >70% concentrated from July to September. Annual evaporation ranges from 1500 to 1800
mm. Three grassland areas abandoned in the same year were selected for the experiment. Details on
soil properties were described in Ai et al. (2017a).
**2.2 Experimental design and soil sampling**
The slopes of the three grassland areas were south-facing (S15°W), northeast-facing (N75°W), and
north-facing (N57°E). The three study areas were selected in September 2014 after consultation with
ARS researchers and reviewing relevant land documents. The basic characteristics are shown in Table 1.
Three replicate 10 × 10 m plots were established at each site (*A. sacrorum* was the dominant plant at
each site). Each plot was first surveyed for latitude, longitude, elevation, slope aspect, and slope
gradient. Three 1 × 1 m quadrats were then randomly set in each plot to characterise the vegetation, e.g.
plant species, coverage, and number. The plants were removed, and the soil strongly adhering to the
roots, i.e. RS, was collected (0–20 cm soil layer). Soil was also sampled from the same layer at
locations approximately 15 cm from the plant roots (i.e. NRS). Each NRS sample was a composite of
subsamples collected at five points (the four corners and the centre of the plot). A total of 18 soil
samples (3 sites × 3 plots per site × 2 soil types) were collected, and each was divided into two
subsamples: one subsample was placed in a cool container, and the other was placed into a cloth bag.
The samples were then taken to the laboratory, and gravel and coarse fragments were removed. The
container samples were homogenised and sieved to 2 mm and were also divided into two subsamples:
one subsample was stored at -80 ℃, and the other was stored at 4 ℃ until analysis. The samples in the
cloth bags were air-dried and sieved to 0.25 and 1 mm prior to analysis.
**Table 1.** Characteristics of the sampling sites.

| Slope aspect | Latitude (°N) | Longitude (°E) | Altitude (m) | Plant community |
|---|---|---|---|---|
| S15°W | 36.85 | 109.31 | 1269 | *A. sacrorum + Bothriochloa ischaemum* |
| N75°W | 36.85 | 109.31 | 1275 | *A. sacrorum + Phragmites australis* |
| N57°E | 36.85 | 109.31 | 1278 | *A. sacrorum + Artemisia capillaries* |

**2.3 Laboratory analysis**
The samples stored at 4 ℃ were used for determining MBC content, basal respiration (BR), and
substrate-induced respiration (SIR). Microbial biomass was measured by chloroform fumigation
(Vance et al., 1987). The soil samples that were fumigated for 24 h at 25.8 ℃ with $CHCl_3$ (ethanol free)
after the fumigation and non-fumigation treatments, and then were extracted with 100 ml of 0.5 M
$K_2SO_4$ by horizontal shaking for 1 h at 200 rpm and then filtered. The amount of $K_2SO_4$-extracted
organic C was determined by a liquiTOCII analyser (Elementar, Hanau, Germany) and MBC content




was calculated using a $k_{EC}$ factor of 0.38 (Vance et al., 1987). BR and SIR were measured by an
infrared gas analyser (QGS-08B, Beijing, China) (Hueso et al., 2011). The metabolic quotient was
calculated as BR per unit MBC (BR/MBC) (Anderson and Domsch, 1993).
The soil stored at -80 ℃ was used for the determination of PLFA contents. The structures of the
microbial communities were determined using a method (Bligh and Dyer, 1959) modified by Bardgett
et al. (1996). Briefly, fatty acids were extracted from 3.0 g of freeze-dried soil using a solution
containing citrate buffer, chloroform, and methanol. The PLFAs were separated from neutral and
glycolipid fatty acids by solid-phase-extraction chromatography. After mild alkaline methanolysis, the
PLFAs were analysed using a gas chromatograph (GC7890A, Agilent Technologies Inc., Wilmington,
USA) equipped with MIDI Sherlock software (Version 4.5; MIDI Inc., Newark, USA). An external
standard of 19:0 methyl ester was used for quantification (Frostegård et al., 1993), and the amounts
were expressed as nmol g$^{-1}$ for dry soil.
According to Zelles (1999), specific PLFA signatures can serve as indicators of specific microbial
groups. Total PLFAs were obtained by summing the contents of all fatty acids detected in each sample.
The classification PLFA was shown in Table 2.
**Table 2.** The characterization of the microbial phospholipid fatty acids.

| Microbial group | Specific PLFA markers |
| --- | --- |
| Gram-positive bacteria | 11:0 anteiso, 12:0 anteiso, 13:0 iso, 13:0 anteiso, 14:0 iso, 14:0 anteiso, 15:0 iso, 15:0 anteiso, 15:1 iso w6c, 15:1 iso w9c, 16:0 iso, 16:0 anteiso, 17:0 iso, 17:0 anteiso, 18:0 iso, 19:0 iso, 19:0 anteiso, 22:0 iso |
| Gram-negative bacteria | 12:1 w4c, 12:1 w8c, 14:1 w5c, 14:1 w8c, 14:1 w9c, 15:1 w5c, 15:1 w7c, 15:1 w8c, 16:1 w7c DMA, 16:1 w7c, 16:1 w9c DMA, 17:0 cyclo w7c, 17:1 w5c, 17:1 w7c, 17:1 w8c, 18:1 w5c, 18:1 w6c, 18:1 w7c, 18:1 w8c, 18:1 w9c, 19:0 cyclo w6c, 19:0 cyclo w7c, 19:1 w6c, 19:1 w8c, 20:1 w6c, 20:1 w9c, 21:1 w3c, 21:1 w5c, 21:1 w6c, 22:1 w3c, 22:1 w5c, 22:1 w6c, 22:1 w8c, 22:1 w9c, 24:1 w9c, 19:0 cyclo 9,10 DMA |
| Fungi | 16:1w5c, 18:2w6c |
| Actinomycetes | 16:0 10-methyl, 17:0 10-methyl, 17:1 w7c 10-methyl, 18:0 10-methyl, 18:1 w7c 10-methyl, 19:1 w7c 10-methyl, 20:0 10-methyl |

The concentrations of soil organic carbon (SOC), total nitrogen (TN), and total phosphorus at the
sites have been reported by Ai et al. (2017a). Soil pH and available phosphorus (SAP), ammonium N
($NH_4$), nitrate N ($NO_3$), water-soluble organic C (WSOC), water-soluble $NH_4$ (WNH$_4$), and
water-soluble $NO_3$ (WNO$_3$) contents were measured as described by Ai et al (2017b).
**2.4 Calculations and statistical analysis**
RE was calculated as: RE=Rs/NRs, where Rs is a microbial property in RS, and NRs is a microbial
property in NRS (Mukhopadhyay et al., 2016). All data were analysed using one-way ANOVAs,
followed by Duncan's tests at a probability level of $P<0.05$ for multiple comparisons. All statistical
analyses were performed using SPSS 20.0 (SPSS Inc., Chicago, USA), and structural equation models
(SEMs) were analysed using the AMOS SPSS expansion pack. Redundancy analysis (RDA) was
performed using CANOCO 5.0 (Biometris, Wageningen, The Netherlands). The graphs were plotted
using SigmaPlot 12.5 (Systat Software, San Jose, USA).
**3 Results**



### 3.1 MBC content, respiration, and BR/MBC

RS MBC content did not differ significantly among the slope aspects, but NRS MBC content in the north-facing slope was higher than those in the south- and northeast-facing slopes (Fig. 1A). RE in the south-facing slope was highest among the slope aspects (Fig. 3). Slope aspect did not affect BR, BR/MBC, or SIR in either RS or NRS (Fig. 1B and Table 3). The RE for BR did not differ significantly among the slope aspects (Fig. 3). The RE for SIR in the south-facing slope was higher than that in the north-facing slope.

### 3.2 Microbial PLFA contents and composition

The microbial PLFA contents in RS differed significantly among the slope aspects. Total PLFA contents in the north- and northeast-facing slopes were 115 and 88% higher, respectively, than that in the south-facing slope (Fig. 2A). Fungal PLFA content did not differ significantly among the slope aspects. Bacterial PLFA content was similar to the trend for total PLFA content, with the lowest content in the south-facing slope. In contrast to total PLFA content, the ratio of fungal PLFA content to bacterial PLFA content (F/B ratio) in the south-facing slope was significantly higher than those in the north- and northeast-facing slopes (Table 3). Both $G^+$ and $G^-$ PLFA contents had trends similar to that of the bacterial PLFA content, with the lowest contents in the south-facing slope (Fig. 2A). The ratio of $G^+$ PLFA content to $G^-$ PLFA content ($G^+/G^-$ ratio) did not differ significantly among the slope aspects (Table 3). With a trend similar to that of $G^-$ PLFA content, actinomycete PLFA content in the north-facing slope was 102% higher than that in the south-facing slope.

**Table 3.** Microbial respiratory quotients (BR/MBC), ratios of fungal PLFA content to bacteria PLFA content (F/B), and ratios of $G^+$ PLFA content to $G^-$ PLFA content ($G^+/G^-$) in the rhizospheric and non-rhizospheric soils.

| Slope aspect | Rhizospheric soil | | | Non-rhizospheric soil | | |
|---|---|---|---|---|---|---|
| | BR/MBC ($10^3$ $h^{-1}$) | F/B ratio | $G^+/G^-$ ratio | BR/MBC ($10^3$ $h^{-1}$) | F/B ratio | $G^+/G^-$ ratio |
| South-facing | 3.03±0.49a | 0.07± 0.00a | 2.16± 0.58a | 2.47±0.52a | 0.07± 0.00a | 1.45± 0.23a |
| North-facing | 2.67±0.41a | 0.03± 0.00b | 1.55± 0.29a | 2.00±0.26a | 0.04± 0.00b | 1.20± 0.10a |
| Northeast-facing | 2.77±0.23a | 0.04± 0.00b | 1.54± 0.16a | 2.47±0.35a | 0.05± 0.00ab | 1.33± 0.06a |

The composition of the NRS PLFA contents also differed significantly among the slope aspects. Total PLFA content in the north-facing slope was 50 and 62% higher than those in the south- and northeast-facing slopes, respectively (Fig. 2B). Bacterial PLFA content had a trend similar to that of total PLFA content, with the highest content in the north-facing slope. Fungal PLFA content in the south- and north-facing slopes was significantly higher than that in the northeast-facing slope. The F/B ratio in the south-facing was substantially higher than that in the north-facing slope (Table 3). $G^-$ PLFA content had a trend similar to that of bacterial PLFA content, and $G^+$ PLFA content did not differ significantly among the slope aspects (Fig. 2B). The $G^+/G^-$ ratio did not differ significantly among the slope aspects (Table 3). Actinomycete PLFA content had a trend similar to that of fungal PLFA content, with the higher contents in the south- and north-facing slopes, which were 49 and 117% higher, respectively, than that in the northeast-facing slope (Fig. 2B).

The REs for total, bacterial, $G^+$, $G^-$, and actinomycete PLFA contents differed significantly among





the slope aspects, but not the RE for fungal PLFA content (Fig. 3). The REs for total, $G^+$, $G^-$, and
bacterial PLFA contents in the northeast-facing slope were highest among the slope aspects. The RE for
actinomycete PLFA content in the northeast-facing slope was highest among the slope aspects.
**3.3 Redundancy analysis (RDA)**
The constrained RDAs indicated that environmental factors affected RS microbial characteristics (Fig.
4A). The total variation was 6.10, and the explanatory variables accounted for 96.8%. The first two
axes (RDA1 and RDA2) explained 89.6% of the total variance, wherein 84.1% was attributed to RDA1
and 5.5% to RDA2. WSOC content was the most significant of the seven environmental factors and
explained 63.6% ($P$=0.006) of the total variance. The slope aspect was the next most significant
environmental variable and explained 62.8% ($P$=0.004), followed by $NH_4$ (58.6%, $P$=0.004), SAP
(45.7%, $P$=0.022), and $WNH_4$ (45.2%, $P$=0.032) contents.
The constrained RDAs indicated that environmental factors affected NRS microbial characteristics
(Fig. 4B). The total variation was 2.97, and the explanatory variables accounted for 94.2%. RDA1 and
RDA2 explained 81.6% of the total variance, 68.3% for RDA1 and 13.3% for RDA2. $WNO_3$ content
was the most significant of the seven environmental factors and explained 34.7% ($P$=0.04) of the total
variance. Slope aspect (33.7%, $P$=0.054) and $WNH_4$ content (32.8%, $P$=0.092) also played important
roles.
**3.4 Path analysis**
The final SEM based on all indices adequately fitted the data to describe the effects of the
environmental factors on RS microbial characteristics ($x^2$=0.506; $P$=0.918; RMSEA, $P$<0.001;
standardised path coefficients are shown in Fig. 5A). The final model accounted for 99% of the
variation in RS WSOC content, with 71% of the variation in bacterial PLFA content, 78% of the
variation in $G^+$ PLFA content, and 72% of the variation in total PLFA content. Slope aspect was
positively correlated with WSOC content ($P$<0.001). WSOC content was negatively correlated with
bacterial PLFA ($P$<0.001), $G^+$ PLFA ($P$<0.001), and total PLFA ($P$<0.001) contents.
All indices adequately fitted the data to describe the effects of the environmental factors on NRS
microbial characteristics ($x^2$=3.222; $P$=0.521; RMSEA, $P$<0.001; standardised path coefficients are
shown in Fig. 5B). The model was able to explain 59% of the variation in $WNH_4$ content, 58% of the
variation in MBC content, 55% of the variation in $G^-$ PLFA content, and 45% of variation in total PLFA
content. Slope aspect was strongly positively correlated with $WNH_4$ content ($P$<0.001). $WNH_4$ content
was strongly negatively correlated with MBC ($P$<0.001), $G^-$ PLFA ($P$<0.05), and total PLFA ($P$<0.05)
contents.
**4 Discussion**
**4.1 MBC, respiration, and BR/MBC**
Soil microbial biomass is closely associated with soil-moisture content (Zhang et al., 2005;Drenovsky
et al., 2010;Ma et al., 2015). The north-facing slope contained more moisture than the south-facing
slope (Sariyildiz et al., 2005), so microbial activity in the north-facing slope was higher than that in the
south-facing slope. NRS MBC content in the north-facing slope was significantly higher than that in
the south-facing slope in our study, supporting our hypothesis 1 and in agreement with other studies
(Huang et al., 2015;Sidari et al., 2008). Carletti et al. (2009), however, reported an opposite trend: soil



MBC was higher in a south-facing slope. This disparity may have been due to the differences in plant
species, soil type, and regional climate (Gilliam et al., 2014). We also found that NRS MBC content in
the northeast-facing slope was lower than that in the north-facing slope, inconsistent with Huang et al.
(2015), whose study area had the same soil and climatic conditions as ours. We speculate that the
different result may mainly due to the different plant species: studied, the effect of shrubland plants
(Huang et al., 2015) on NRS may be different from grassland plants (ours). As plant shade can affect
soil microbial activity (Blok et al., 2010), different shading can cause different MBC contents.

Plant roots release a high amount of exudates, such as sugars, amino acids, organic acids,
hormones, and enzymes (Zhang et al., 2012;Grayston et al., 1997). In contrast, soil with a low amount
of shading is prone to desiccation (Wang et al., 2008), and the quantity of exudates released by plant
roots is low, which may lead to lower activities of the microorganisms. RS MBC content therefore
should be higher than NRS MBC content, consistent with our results. In our study, the RE for MBC
content in the south-facing slope was significantly higher than that in the north-facing slope.
South-facing slopes in the Northern Hemisphere receive more sunlight, which would have a greater
impact on the soil micro-environmental light than that in north-facing slopes between RS and NRS.
The RDA and path analysis found that NRS WSOC, $WNO_3$, and $WNH_4$ contents were well correlated
with MBC content (Figs. 4B and 5B), supporting our hypothesis 2 and in agreement with other studies
(Haynes, 2000;Huang et al., 2014).

Neither RS nor NRS BR, SIR, and BR/MBC differed significantly among the slope aspects,
indicating that the actual microbial activities, potential microbial activities, and bioenergetic status of
the microbial biomass (Nannipieri et al., 1990;Wardle, 1995;Sinha et al., 2009) were similar among the
slope aspects in the study area. The RE of SIR in the south-facing slope was 96% higher than that in
the north-facing slope, indicating that the effect of slope aspect on the RS and NRS SIRs was more
evident in the south-facing slope than that in the north-facing slope, even though the influence of slope
aspect on SIR was not significant either in RS or NRS.
**4.2 PLFA contents and composition**
**4.2.1 Fungal and bacterial PLFA contents and composition**
NRS fungal PLFA content in the northeast-facing slope was lower than those in the south- and
north-facing slopes, however, RS fungal PLFA content did not differ significantly among the slope
aspects. Previous studies have reported different results: Huang et al. (2015) and Gilliam et al. (2014)
found that slope aspect significantly affected the fungal community, and fungal abundance was lower
in north-facing slope; Bardelli et al. (2017) found that fungal abundance did not differ significantly
between north- and south-facing slopes. These different results may due to the differences in plant
species (e.g. herbs vs shrubs), soil conditions, climate, and research methods (Gilliam et al., 2014). The
different responses of RS and NRS fungal PLFA contents meant that rhizospheres could form an
environment that negates the effect of slope aspect on fungal communities more than in
non-rhizospheric zones. SOC and TN can supply the microbial biomass with enough C, N, and energy
resources to support microbial growth (Jia et al., 2005), so the solubility of SOC (WSOC) and TN
($WNO_3$, $WNH_4$) would be closely associated with the fungal community. The RDA showed that NRS
WSOC and $WNO_3$ were well correlated with fungal PLFA content (Fig. 4B), supporting our hypothesis
2 and agreed by previous studies (Haynes, 2000;Nilsson et al., 2005;Huang et al., 2014).

The effect of slope aspect on PLFA content differed between bacteria and fungi. Both RS and NRS
bacterial PLFA contents in the south-facing slope were lower than those in the north-facing slope,



suggesting more soil moisture in the north-facing slope suitable for the growth of bacteria, in agreement with some studies (Huang et al., 2015;Ascher et al., 2012) but not others (Gilliam et al., 2014;Bardelli et al., 2017). The effect of slope aspect on the bacterial community would therefore become significant due to the plant species, soil type, and climatic conditions. The RE for bacterial PLFA content in the northeast-facing slope was significantly higher than that in the south-facing slope, indicating that the environmental conditions of the rhizosphere helped the bacterial community to resist environmental pressure. The RDA indicated that the RS $WNH_4$, WSOC, and SAP contents were well correlated with the bacterial PLFA content, and the NRS $WNH_4$ content was well correlated with the bacterial PLFA content (Fig. 4A, B). The path analysis indicated that RS WSOC content was the main factor influencing the bacterial PLFA content and mainly affected the $G^+$ PLFA content (Fig. 5A), in agreement with another study (Fierer et al., 2003), but the NRS $WNH_4$ content mainly affected the $G^-$ PLFA content (Fig. 5B). These results indicated that the RS and NRS bacterial PLFA contents were affected by different soil nutrient factors.

Soil moisture is an important environmental factor affecting the composition of microbial communities, the higher amounts of soil moisture in north-facing slopes (Sariyildiz et al., 2005) can lead to lower F/B ratios (Brockett et al., 2012;Drenovsky et al., 2010;Ma et al., 2015). In this paper, the F/B ratio was highest in the north-facing slope and lowest in the south-facing slope for both RS and NRS, consistent with previous studies (Huang et al., 2015;Gilliam et al., 2014). The higher amount of soil moisture in the north-facing slope would reduce soil aeration, lower oxygen levels would create an environment favourable for facultative and obligate anaerobic bacteria (Drenovsky et al., 2004). Drought stress in the south-facing slope would likely facilitate the survival of fungi, because soil fungi rely on more aerobic conditions and are more tolerant of drought due to their filamentous nature (Zhang et al., 2005).

The significant difference in the RE for bacterial PLFA content was not obvious for fungal PLFA content, so the RE was much weaker in the fungal than the bacterial community, consistent with Buyer et al. (2002). These results indicated that RE had a large effect on the structures of the fungal and bacterial communities. RE was significantly affected by slope aspect for both the $G^+$ and $G^-$ PLFA contents, and their REs were consistent with the RE of the total bacterial PLFA content. The $G^+/G^-$ ratio can indicate the dominance of bacteria in soil microbial communities (Tscherko et al., 2004;Zhang et al., 2015). Neither the RS nor the NRS $G^+/G^-$ ratio was affected by slope aspect, indicating that slope aspect did not significantly affect the dominant bacterial community in either RS or NRS.

### 4.2.2 Actinomycete and total PLFA contents

RS and NRS actinomycete PLFA contents were significantly affected by slope aspect, supporting our hypothesis 1. Actinomycetes and $G^+$ bacteria have similar life habits, so wetter soils are more enriched in actinomycetes (Zhang et al., 2005;Drenovsky et al., 2010;Ma et al., 2015). RS actinomycete PLFA content in the north-facing slope was therefore higher than that in the south-facing slope, and the northeast-facing slope had more moderate growth conditions for actinomycetes compared with the north-facing and south-facing slopes. NRS actinomycete PLFA content, however, was lower in the northeast-facing slope than that in the south-facing slope. This difference may have been due to RE, because RE in the northeast-facing slope was significantly higher than those in the other slopes. RE will affect soil nutrients more in RS than NRS (Zhang et al., 2012;Grayston et al., 1997). The RDA indicated that the RS but not NRS actinomycete PLFA content was well correlated with WSOC and $WNH_4$ contents, supporting our hypothesis 2 (Fig. 4A, B).



The $G^+$ and actinomycete PLFA contents accounted for more than 50% of total PLFA content in
both RS (57–59%) and NRS (54–58%), so the distribution of total PLFA content in our study area
depended mainly on the $G^+$ and actinomycete PLFA contents. Drier soils tend to be more enriched in $G^-$
bacteria and fungi, whereas wetter soils tend to be more enriched in $G^+$ bacteria and actinomycetes
(Zhang et al., 2005;Drenovsky et al., 2010;Ma et al., 2015), so total PLFA contents in both RS and
NRS were highest in the north-facing slope. Total PLFA content, however, was higher in the
northeast-facing slope than that in the south-facing slope for RS, and did not differ significantly
between the northeast-facing and south-facing slopes in NRS. These differences in total PLFA content
between RS and NRS may have been mostly due to RE. The shading by herbs in the northeast-facing
slope may make RS was suitable for microbial life as in the north-facing slope, whereas NRS in the
northeast-facing slope was not suitable for microbial life as in the south-facing slope without plant
shading. The path analysis indicated that WSOC content had a significant effect on RS total PLFA
content and that $WNH_4$ content had a significant effect on NRS total PLFA content (Fig. 5A, B), as
expected (Haynes, 2000;Huang et al., 2014;Nilsson et al., 2005). These results supported our
hypothesis1and 2, but hypothesis1were inconsistent with Huang et al. (2015) who found a significantly
higher total PLFA content in the south-facing slope than in other slopes. RE may be one of the main
reasons, because shrub shading (Huang et al., 2015) clearly differs from herb shading (our study),
which could by caused a different RE (Blok et al., 2010).
**5 Conclusions**
This study provides experimental evidence that slope aspect can markedly but differentially affect
MBC and PLFA contents in RS and NRS, and the different influences can produce an evident RE; the
RE for most microbial properties was higher in the northeast-facing slope. WSOC content was well
correlated with RS microbial properties, and $WNH_4$ content was well correlated with NRS microbial
properties, likely due to RE. Studies of the influence of slope aspect on soil microbial communities
should therefore consider REs. This study provides new insights into the influences of topographic
factors affecting the mechanisms driving the structure of microbial communities in a
micro-ecosystemic environment. Further field investigation on different plant species, however, is
needed to determine the role of RE under the effect of slope aspect in micro-ecosystemic environments.
**Author contributions**
GbL and SX provided research ideas and designed the experiments. They were also responsible for the
revision of the paper. SX, ZmA, JyZ and HfL participated in the soil sample collection, ZmA, JyZ and
HfL contributed to the soil analysis. ZmA analyzed the data and wrote the paper.

*Acknowledgements.* We thank the National Natural Science Foundation of China (41371510, 41671513,
41471438) and the West Young Scholars Project of The Chinese Academy of Sciences (XAB2015A05)
for funding this work.

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



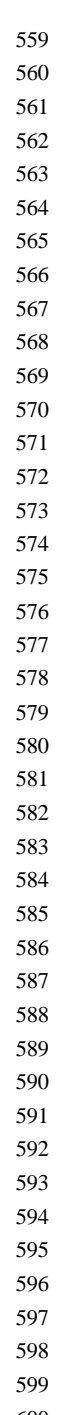

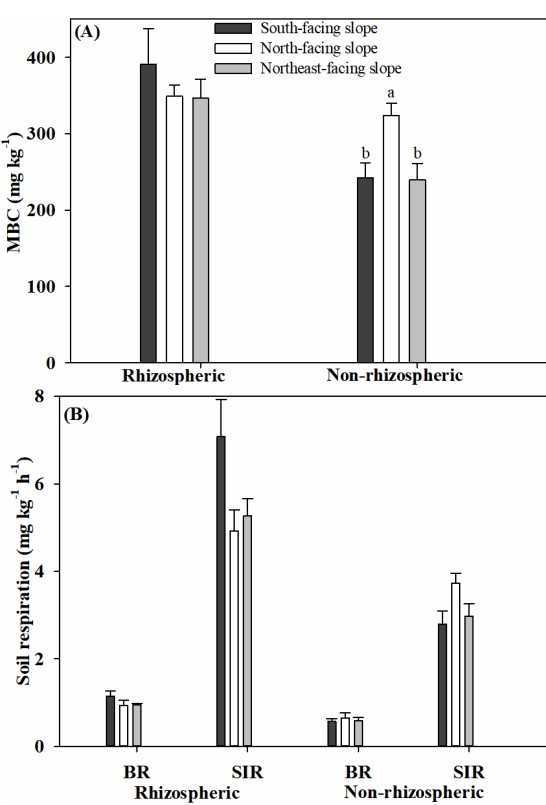

**Fig. 1** Microbial biomass carbon (MBC) content, basal respiration (BR), and substrate-induced respiration (SIR) in the rhizospheric and non-rhizospheric soils. Error bars are standard errors (n=3). Different letters above the bars indicate significant differences at $P$=0.05.

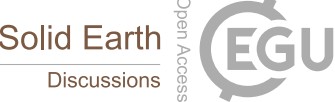

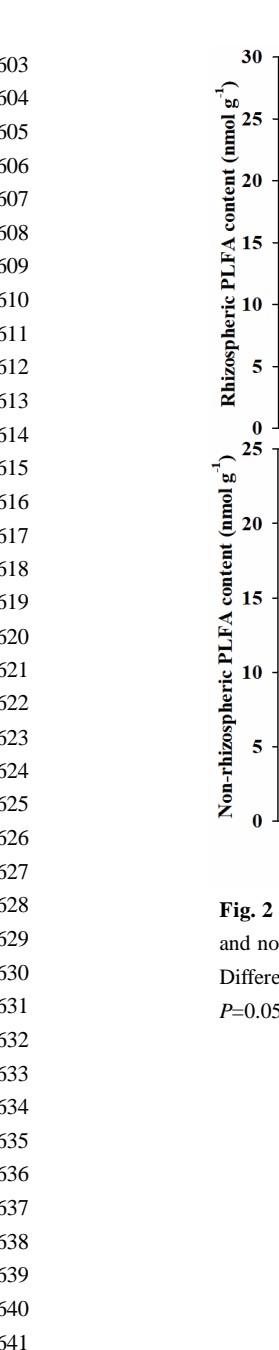

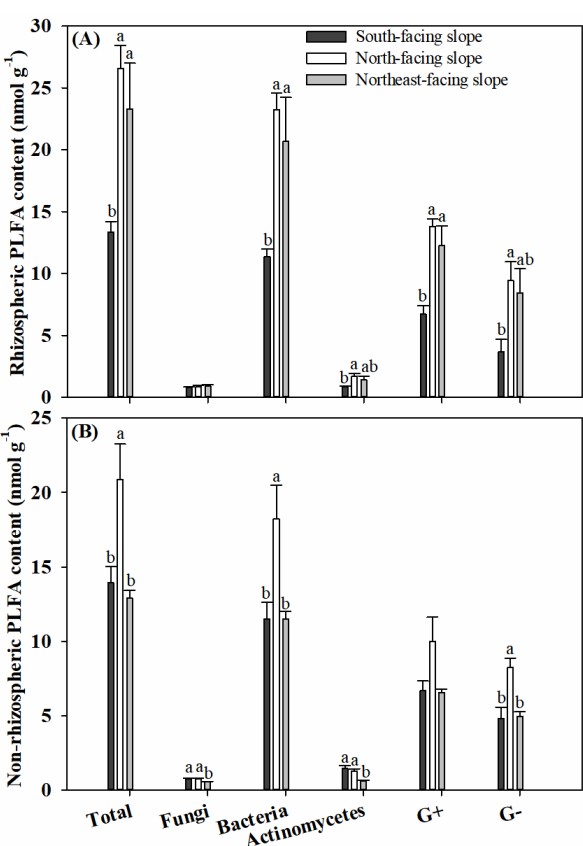

**Fig. 2** Effects of slope aspect on PLFA contents in rhizospheric and non-rhizospheric soils. Error bars are standard errors (n=3). Different letters above the bars indicate significant differences at $P$=0.05.

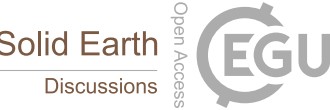

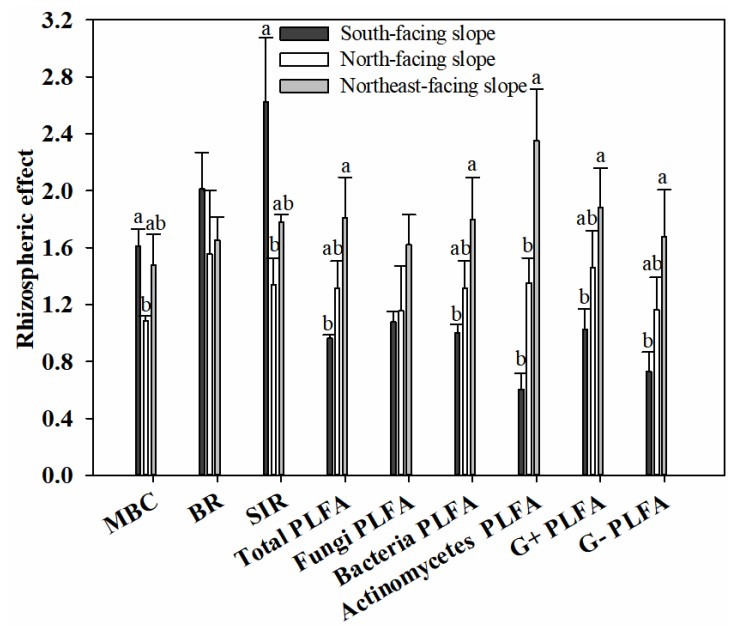

**Fig. 3** The rhizospheric effect of microbial biomass carbon, basal respiration, substrate-induced respiration, and PLFA contents. Error bars are standard errors (n=3). Different letters above the bars indicate significant differences at *P*=0.05.





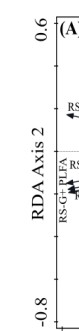
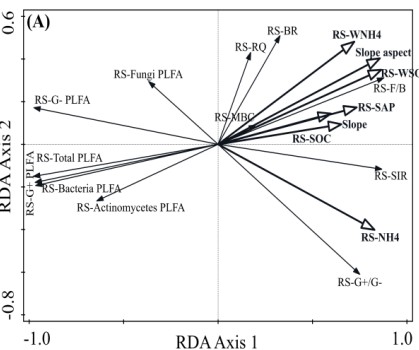
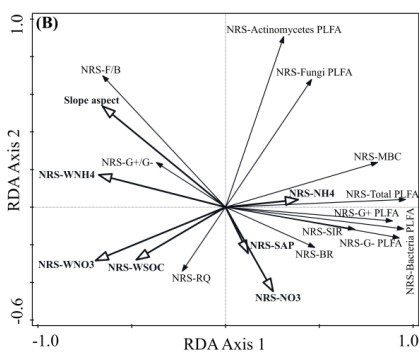


**Fig. 4** Bidimensional graph for a redundancy analysis of the relationships between microbial properties and environmental factors in the rhizospheric (A) and non-rhizospheric (B) soils.


Note: RS, rhizospheric soil; NRS, non-rhizospheric soil; RQ, respiratory quotient; F/B, F/B ratio; G$^+$/G$^-$, G$^+$/G$^-$ ratio.






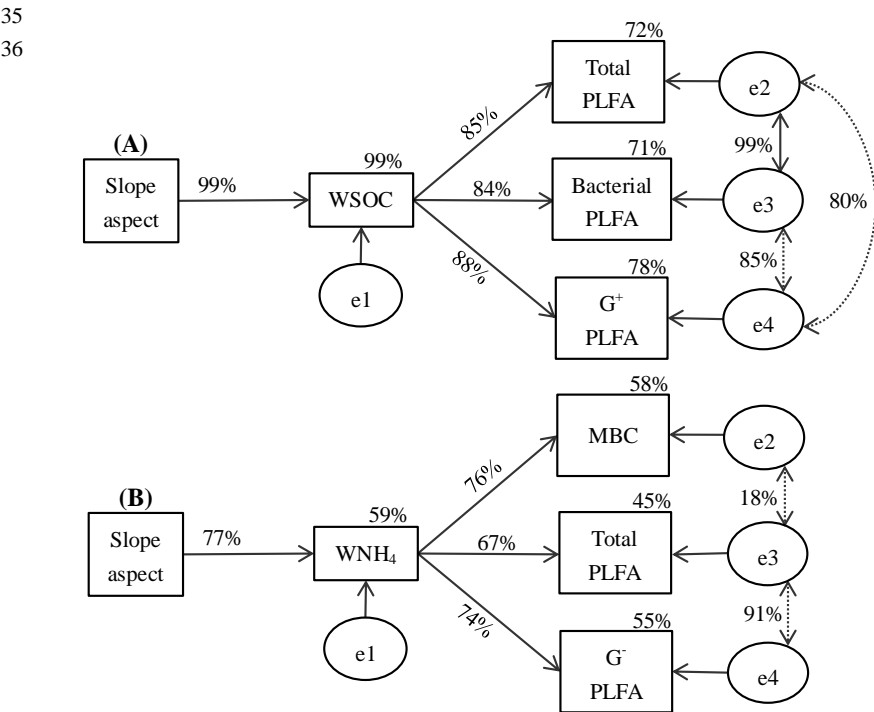

**Fig. 5** Structural equation models of the effect of slope aspect on microbial properties in the rhizospheric (A) and non-rhizospheric (B) soils. Numbers on the arrows are standardised path coefficients (equivalent to correlation coefficients). Solid lines indicate significant standardised path coefficients ($P$ <0.05). Circles indicate error terms (e1–e4). Percentages near the endogenous variables indicate the variance explained by the model.