# Peer review of "Influence of slope aspect on the microbial properties of rhizospheric and non-rhizospheric soil on the Loess Plateau, China"

_Solid Earth, 2017_

## Referee Comment (RC1) · Anonymous Referee #1 · 26 Jan 2018

General comments The manuscript investigated the influence of slope aspects (south-facing, north-facing, and northeast-facing slopes, all with Artemisia sacrorum as the dominant species) on RS and NRS microbial biomass carbon (MBC) and phospholipid fatty acid (PLFA) contents, and the rhizospheric effect (RE) of various microbial indices. Using redundancy analysis (RDA) and path analysis, the authors quantified the driving factors controlling the rhizospheric soil (RS) and non-rhizospheric soil (NRS) microbial properties. I think the study is quite interesting, however, I think a total of 18 soil samples (3 sites × 3 plots per site × 2 soil types) were not enough for redundancy analysis, and the number of soil samples is not enough for evaluating the influence of slope aspect on the microbial properties of rhizospheric and non-rhizospheric soil, it

will be perfect to take more soil samples at different slope gradient. Overall I think the paper will be of interest to soil readers. However, it needs considerable work, though, before it's ready for publication in the field. That work includes better organization of manuscript (especially Introduction section) and more sampling for the results. More importantly, The English in the paper needs considerable editing. Here are some more specific suggestions to improve the manuscript in a revised version. Specific comments Lines 35: the first word "Slope aspect is an important topographic factor," is not incomplete. Line 55: the sentence "the angle between the ground and wind direction, which is defined as the orientation faced by a slope" can be deleted. Line 61-65: the sentences move to the behind of the first sentence of this paragraph, or suggest delete. Line 64- 69: the two sentences don't connect well. Line 68-72: suggest summarize these sentences. Line 70: unclear statement. Line 108-109: unclear statement. Line 121-122: unclear statement. Line 185: suggest "Impacts of slope aspects on. . .." Line 192: suggest "Impacts of slope aspects on. . .." Line 186-188: place the "Fig" in order. Line 232-235: unclear statement. Line 239, 245: "standardised path coefficients are shown in" suggest delete. Line 702: please give the full name of WSOC, SOC, SAP. . .

---

## Referee Comment (RC2) · Anonymous Referee #2 · 7 Feb 2018

This paper is interesting and urgent in sense of microbial soil properties. The authors have investigated effect of slope aspect (south-facing, north-facing, and northeast-facing slopes) on the microbial properties of grassland rhizospheric and non-rhizospheric soils. The subject site was Ansai Research Station on the Loess Plateau in China. The authors have analyzed microbial biomass carbon, basal respiration, substrate-induced respiration, phospholipid fatty acid contents and the rhizospheric effect. Moreover, statistics was used in the research (redundancy analysis and path analysis). The paper has a good organization, but the language is partly insufficient. It should be helpful to get the revision of a native English speaking person. I suggest several comments to improve the manuscript: 1. There is no explanation of the obtained results in the abstract - just briefly, in 1-2 sentences. 2. Please, pay attention to articles using in the text. 3. Line 57: "Solar radiation influences ON (?) ecologically critical factors..." 4. Line 61: "which receiVE the more solar radiation..." 5. In my opinion, it is worth to include to the introduction part the latest papers devoted to studied topic (2015-2017). 6. Throughout the manuscript: please, don't use the pronoun "We" (for example, lines 114,116,117, 260, 262) – sentences should be impersonal. 7. Why did you investigate only 3 variants - south-facing, north-facing, and northeast-facing slopes? Why didn't you investigate west slopes? Was is the idea? 8. Hypotheses (1 and 2) should be written not in Future tense (would), but in Present. 9. Lines 121-122 – it is unclear – could you rephrase? 10. Line 127: "annual temperature OF the study area is 8.8 °C,..." 11. Section 2.1 Study site: Please provide the short review of natural soil and vegetation diversity in the region, close to the study plots, soil classification with references. Please, provide the detailed scheme/map of sampling sites putting sampling plots on it. 12. Line 129: Please, write a little bit more detail about abandoned areas – what happened and why, when it was and so on.. 13. Please, specify distance between sampling plots within sampling site and between them. 14. Are 18 soil samples enough to do the statistics that you have done? 15. Section 2.3 Laboratory analysis: What are units of measure for MBC content, BR, SIR and metabolic content? 16. Line 158: Please, provide a detail formula for calculation a metabolic content (in order to understand units of measure). 17. Lines 156-157: Please, provide more detail description of BR and SIR analysis (like MBC). 18. Lines 177-178: Please, write more detailed how RE was calculated, or give an example of calculation. 19. Lines 187-188: it is unclear - RE in the south-facing slope was highest among the slope aspects – it is not for all studied properties. 20. Line 189: "...or SIR in either RS or NRS..." – for SIR this statement is not true according to the Fig. 1B. 21. Line 208: "...Total PLFA content in the north-facing slope was 50 and 62% higher than those..." – I suppose that 50 and 62% are incorrect – according to the Fig. 2B. 22. Line 213: "...G+ PLFA content did not differ significantly among the slope aspects (Fig. 2B)." – according to the Fig. 2B, it is not true. 23. Line 216: "...which were 49 and 117% higher..." – I suppose

that 49 and 117% are incorrect – according to the Fig. 2B. 24. Lines 219-221: these two sentences can be summarized. 25. Lines 173-175: you have measured several characteristics (pH, SAP, WSOC, and so on), but in results section there are no data. Could you, please, include a table or figure with these characteristics? 26. Line 275: "...supporting our hypothesis 2 ..." – could you write it more precisely. 27. Line 278: what do you mean – bioenergetic status – could you explain in your opinion? 28. Line 291: "These different results may BE due to the differences" 29. There is not so much discussion about north-east slopes in the text – could you add it? 30. Lines 318-319: this statement is not correct according to table 3. 31. Line 315-324. If soil moisture is an important environmental factor affecting the composition of microbial communities, I suppose you should add your data of soil moisture to manuscript. 32. Sections 4.2.1 and 4.2.2 – please, analyze data of G+ and G- PLFA contents either in section 4.2.1 or 4.2.2. 33. Line 339-340: "NRS actinomycete PLFA content, however, was lower in the northeast-facing slope than that in the south-facing slope." - but also - than in the north-facing slope, or not? 34. Lines 347-348: "Drier soils tend to be more enriched in G- bacteria and fungi,..." – it is not correct according to fig. 2. 35. Line 408: the surname SCHAEPMAN‐STRUB should be written in lower case. 36. Journal names should be abbreviated according to the ISI Journal Title Abbreviations Index (according to Manuscript preparation guidelines for authors). 37. Please, add DOI to references. 38. The quality of Figures 2 and 3 is very poor. Nothing is clear at the picture. Please make columns larger and readable. Could you explain what different letters above the bars (a, b, ab) mean? And when there are no letters – what does it mean? (fig. 1-3). 39. Figure 4: RQ (respiratory quotient) – what is it? Please, make this figure larger and readable.

Nevertheless, I found this paper of good quality and after correcting it can be publish.
* * *

---

## Author Comment (AC1) · 11 Feb 2018

Dear Reviewer, We are very glad to receive the comments to our manuscript se-2017-137, entitled "Influence of slope aspect on the microbial properties of rhizospheric and non-rhizospheric soil on the Loess Plateau, China". The comments from the reviewer are very helpful for revising and improving our manuscript, as well as hold great guiding significance to our researches. We take all of these comments into account in preparing the revised manuscript. We believe that the manuscript has been improved satisfactorily and hope it will be accepted for publication in Solid Earth. We thank again the reviewer for the works that you have done for our paper. If you require any further

information, please contact with us at any times.

General comments The manuscript investigated the influence of slope aspects (south-facing, north-facing, and northeast-facing slopes, all with Artemisia sacrorum as the dominant species) on RS and NRS microbial biomass carbon (MBC) and phospholipid fatty acid (PLFA) contents, and the rhizospheric effect (RE) of various microbial indices. Using redundancy analysis (RDA) and path analysis, the authors quantified the driving factors controlling the rhizospheric soil (RS) and non-rhizospheric soil (NRS) microbial properties. I think the study is quite interesting, however, I think a total of 18 soil samples (3 sites 3 plots per site 2 soil types) were not enough for redundancy analysis, and the number of soil samples is not enough for evaluating the influence of slope aspect on the microbial properties of rhizospheric and non-rhizospheric soil, it will be perfect to take more soil samples at different slope gradient. Overall I think the paper will be of interest to soil readers. However, it needs considerable work, though, before it's ready for publication in the field. That work includes better organization of manuscript (especially Introduction section) and more sampling for the results. More importantly, The English in the paper needs considerable editing. Here are some more specific suggestions to improve the manuscript in a revised version. R: We are very grateful to the reviewers for the recognition, and totally agree with the reviewer's comments. As the reviewer's comments, the plots in our paper were not very enough to explain the effect of slope aspect on the microbial properties of rhizospheric and non-rhizospheric soil, in a certain extent. However, the purpose of this paper was to emphasize that the slope aspect plays a key role in the study of rhizosphere microorganism, and to provide some new ideas for scientific research in a related field. In addition, the field survey in this paper was conducted in September 2014, so the plant community in the study area has now changed, looking for a new gradient of slope aspect was very difficult (other geographical factors as consistent as possible with the previous ones.) in the study area. Actually, we also tried to investigate more slope aspects in the investigation at 2014, but under the conditions of the same site conditions and plant species, the three slope aspects have been chosen in the paper: same site conditions, same

dominant species, and geographical proximity. The location of the three sites were marked in another article (Ai, Z. M., He, L. R., Xin, Q., et al. Slope aspect affects the non-structural carbohydrates and C: N: P stoichiometry of Artemisia sacrorum on the Loess Plateau in China, Catena, 152, 9-17, 2017 ). Figure was shown in the below, Fig. 1. To summarize, we hope that the reviewer consider the field test conditions, and give us a chance to express the standpoint in this article. The paper has been re organized as suggested by the reviewer, especially the introduction part. This paper has been re-edited by the English editor. We are very grateful again to the reviewer for sincere comments, and these suggestions are very important to our future research work.

Specific comments Lines 35: the first word "Slope aspect is an important topographic factor," is not incomplete. R: Considering the reviewer's suggestion, we have changed this sentence into "Slope aspect is an important topographic factor in the micro-ecosystemic environment". Lines 35

Line 55: the sentence "the angle between the ground and wind direction, which is de-fined as the orientation faced by a slope" can be deleted. R: After carefully considered the reviewer's suggestions, we have deleted this sentence. Line 56-57

Line 61-65: the sentences move to the behind of the first sentence of this paragraph, or suggest delete. R: As suggested by the reviewer, we have already put this sentence to behind of the first sentence of this paragraph. Line 59-63

Line 64- 69: the two sentences don't connect well. R: We totally agree with this com-ment, and apologize for our carelessness. We have added the new sentences between the "two sentences": "Previous research indicated that slope aspect markedly affects soil and microbiological properties in micro-ecosystemic environments. The results of studies on the impact of slope aspect on the microbiological properties, however, are not consistent". Line 68-70

Line 68-72: suggest summarize these sentences. R: As suggested by the reviewer, we

have added a sentence in the manuscript: "The influence of slope aspect on microbial characteristics has obviously been variable in these studies, and the differences may be caused by the differences in plant species (trees vs shrubs), soil properties, climatic conditions, and research methods". Line 76-78

Line 70: unclear statement. R: We have revised this sentence to make it clear: "other studies have found that the MBC, fungal, and total phospholipid fatty acid (PLFA) contents in the south-facing slopes were significantly higher than those in north-facing slopes". Line 72-74

Line 108-109: unclear statement. R: We have rewritten this sentence to make it clear: "Under the conditions of different slope aspects, the effect of the main soil nutrient factors on RS and NRS microbial communities on local micro-ecosystemic environments, however, remains unclear". Line 112-114

Line 121-122: unclear statement. R: We have rewritten this sentence to make it clear: "soil carbon (C) and nitrogen (N) are the main soil nutrient factors that affect RS and NRS microbial communities under different slope aspects". Line 126-128

Line 185: suggest "Impacts of slope aspects on. . ." Line 192: suggest "Impacts of slope aspects on. . ." R: As suggested by the reviewer, we have revised those in the manuscript. Line 191, 198

Line 186-188: place the "Fig" in order. R: We have placed the "Fig" in order in the manuscript. Line 192-194

Line 232-235: unclear statement. R: We have changed this sentence into "Among the seven environmental factors, WNO3 content was the most significant and explained 34.7% (P=0.04) of the total variance". Line 238-240

Line 239, 245: "standardised path coefficients are shown in" suggest delete. R: We agree with this suggestion and have deleted the "standardised path coefficients are shown in" in the manuscript. Line 243, 250

Line 702: please give the full name of WSOC, SOC, SAP... R: We are very sorry for our negligence and have added the full name of WSOC, SOC, SAP ... in the manuscript. Line 705

Please also note the supplement to this comment:
https://www.solid-earth-discuss.net/se-2017-137/se-2017-137-AC1-supplement.pdf
* * *
[Figure]

[Figure]

Fig. 1. The location of the experiment sites. ARSSWCCAS = Ansai Research Station of Soil and Water Conservation of the Chinese Academy of Sciences.

**Fig. 1.**

**Supplement:**

Feb 11, 2018

Solid Earth

Dear Reviewer,

This letter certifies that I have edited the language of the manuscript entitled "Influence of slope aspect on the microbial properties of rhizospheric and non-rhizospheric soil on the Loess Plateau, China". The language of the manuscript should meet your standards.

Sincerely,

Dr. William Blackhall

Editor, Global Biological Editing

editor@globalbiologicalediting.com

www.globalbiologicalediting.com

---

## Author Comment (AC2) · 17 Feb 2018

Dear Reviewer, We are very glad to receive the comments to our manuscript se-2017-137, entitled "Influence of slope aspect on the microbial properties of rhizospheric and non-rhizospheric soil on the Loess Plateau, China". The comments from the reviewer are very helpful for revising and improving our manuscript, as well as hold great guiding significance to our researches. We take all of these comments into account in preparing the revised manuscript. We believe that the manuscript has been improved satisfactorily and hope it will be accepted for publication in Solid Earth. We thank again the reviewer for the works that you have done for our paper. If you require any further

information, please contact with us at any times.

Anonymous Referee #2 This paper is interesting and urgent in sense of microbial soil properties. The authors have investigated effect of slope aspect (south-facing, north-facing, and northeast-facing slopes) on the microbial properties of grassland rhizospheric and nonrhizospheric soils. The subject site was Ansai Research Station on the Loess Plateau in China. The authors have analyzed microbial biomass carbon, basal respiration, substrate-induced respiration, phospholipid fatty acid contents and the rhizospheric effect. Moreover, statistics was used in the research (redundancy analysis and path analysis). The paper has a good organization, but the language is partly insufficient. It should be helpful to get the revision of a native English speaking person. R: We are very grateful to the reviewer for the recognition and totally agree with the reviewer's comments. As suggested by the reviewer, the paper has been re-edited by the English editor.

Specific comments I suggest several comments to improve the manuscript: 1. There is no explanation of the obtained results in the abstract - just briefly, in 1-2 sentences. R: Considering the reviewer's suggestion, we have added a sentence in the abstract in the manuscript: "Soil samples were collected from the three slope aspects, including rhizospheric and non-rhizospheric region, and analyzed to determine the related various microbial indices. The results showed. . .." Line 41-42 2. Please, pay attention to articles using in the text. R: Thanks for the reviewer's reminding, we have paid more attention to them. 3. Line 57: "Solar radiation influences ON (?) ecologically critical factors. . ." R: We have a confirmation that there is no "on" here after we communicated with our English editor. When "influence" is a noun, there will be an "on" on the back of "influence", and no "on" when "influence" is a verb. Line 59 4. Line 61: "which receive the more solar radiation. . ." R: In accordance with the reviewer's suggestion, we have changed "receives" to "receive" in the manuscript. Line 62 5. In my opinion, it is worth to include to the introduction part the latest papers devoted to studied topic (2015-2017). R: We agree with the reviewer's viewpoint. In order to control the

number of references in the manuscript, three new references have been added in the manuscript: "Slope aspect can therefore substantially affect soil-moisture content, water budget, and soil temperatures (Dearborn and Danby, 2017)" and "The effect of slope aspect on basic soil properties (pH, bulk density, and texture), nutrient (carbon, nitrogen, and phosphorus) contents, microbial biomass, and enzymatic activities have been studied (Bardelli et al., 2017; Liu et al., 2017)." Line 66, 70 6. Throughout the manuscript: please, don't use the pronoun "We" (for example, lines 114,116,117, 260, 262) – sentences should be impersonal. R: We are very grateful to the reviewer's suggestion, and have deleted all the "We" in the manuscript. 7. Why did you investigate only 3 variants - south-facing, north-facing, and northeast-facing slopes? Why didn't you investigate west slopes? Was is the idea? R: The suggestion is very meaningful. Actually, we also tried to investigate more slope aspects in the investigation at 2014 in the study area, but under the conditions of the same site conditions and plant species, the three slope aspects in the paper have met the conditions: same site conditions, same dominant species, and geographical proximity. 8. Hypotheses (1 and 2) should be written not in Future tense (would), but in Present. R: Considering the reviewer's suggestion, we have changed the hypotheses to "(1) slope aspect significantly but differentially affects the MBC, total, fungal, bacterial, and actinomycete PLFA contents and their REs; and (2) soil carbon (C) and nitrogen (N) are the main soil nutrient factors that affect RS and NRS microbial communities under different slope aspects." Line 127-130 9. Lines 121-122 – it is unclear – could you rephrase? R: We have rewritten this sentence to make it clear: "soil carbon (C) and nitrogen (N) are the main soil nutrient factors that affect RS and NRS microbial communities under different slope aspects". Line 129-130 10. Line 127: "annual temperature OF the study area is 8.8 °C,..." R: We have revised it in the manuscript: "The mean annual temperature of the study area is 8.8 °C, ...". Line 134-135 11. Section 2.1 Study site: Please provide the short review of natural soil and vegetation diversity in the region, close to the study plots, soil classification with references. Please, provide the detailed scheme/map of sampling sites putting sampling plots on it. R: This suggestion is very meaningful to

improve the manuscript. We have described the soil, vegetation, and map of sampling sites in another paper (Ai, Z. M., He, L. R., Xin, Q., et al. Slope aspect affects the non-structural carbohydrates and C: N: P stoichiometry of Artemisia sacrorum on the Loess Plateau in China, Catena, 152, 9-17, 2017 ). In order to reduce the text and length of this manuscript, we did not describe these again, and have added two sentences: "The main vegetation in the region includes woods such as Robinia pseudoacacia and Platycladus orientalis; shrubs such as Caragana korshinskii, Hippophae rhamnoides, Syzygium aromaticum, and Ostryopsis davidiana; and herbage such as Artemisia sacrorum, Bothriochloa ischcemum, Setaria viridis, Artemisia giraldii, and Artemisia capillaris." and "Details of the soil properties, and map of sampling sites were described by Ai et al. (2017)". Certainly, if the reviewer thinks it is necessary to describe these in this manuscript, we can add them. Line 141-146 12. Line 129: Please, write a little bit more detail about abandoned areas – what happened and why, when it was and so on.. R: As suggested by the reviewer, we have added a few sentences in the manuscript: "To control soil erosion and improve the ecological environment, the Chinese government has implemented the policy of converting sloping cropland to grassland in the region in 1990s. Synchronously, restoration of the local grassland mainly dependent on abandoned farmland. In order to study the effect of slope aspect on the soil microbial community in the restored grassland, three grassland areas abandoned in the same year were selected for the experiment." Line 137-141 13. Please, specify distance between sampling plots within sampling site and between them. R: As suggested by the reviewer, we have added the sentence "The distance between sampling plots within sampling site was not less than 20 m" in the manuscript. Line 149 14. Are 18 soil samples enough to do the statistics that you have done? R: This is a very good question. After consulting the researchers in the relevant major, all the co-authors of this manuscript believed that 18 soil samples are enough to do the statistics in a certain extent. 15. Section 2.3 Laboratory analysis: What are units of measure for MBC content, BR, SIR and metabolic content? R: Considering the reviewer's suggestion, we have added the units of measure for MBC content "(mg kg-1)", BR "(mg kg-1 h-1)", SIR

"(mg kg-1 h-1)" and metabolic content "(103 h-1)" in the manuscript. Line 163-164 16. Line 158: Please, provide a detail formula for calculation a metabolic content (in order to understand units of measure). R: As suggested by the reviewer, we have added the formula in the manuscript: "metabolic quotient=103×BR/MBC=103×(mg kg-1 h-1)/(mg kg-1)". Line 197 17. Lines 156-157: Please, provide more detail description of BR and SIR analysis (like MBC). R: As suggested by the reviewer, we have added the determination method of BR and SIR in the manuscript: "The soil BR was estimated by measuring the $CO_2$ evolution from 10.0 g of field fresh soils. The homogenized soil samples were first placed in a polyethylene bottle with rubber stopper (the soil water content was adjusted to 50% of field water-holding capacity). The polyethylene bottle was then incubated at 28 °C for 2 h, and the $CO_2$ evolution was measured by an infrared gas analyser (QGS-08B, Beijing, China) (Hueso et al., 2011). Soil SIR was determined using the same method as for BR but with the addition of 0.06 g glucose to the soil, after the glucose and soil were fully compounded, they were then incubated at 28 °C for 1 h." Line 169-176 18. Lines 177-178: Please, write more detailed how RE was calculated, or give an example of calculation. R: As suggested by the reviewer, we have added an example of calculation of RE: "For example, the RE for MBC: RE=RS MBC/NRS MBC=(mg kg-1)/(mg kg-1)". Line 199-200 19. Lines 187-188: it is unclear - RE in the south-facing slope was highest among the slope aspects – it is not for all studied properties. R: We are very sorry for our negligence and have corrected it in our manuscript: "The RE for MBC in the south-facing slope was highest among the slope aspects". Line 210 20. Line 189: ". . . or SIR in either RS or NRS . . ." – for SIR this statement is not true according to the Fig. 1B. R: The statistical results of SIR in either RS or NRS indicated that there were no significant difference among different slope aspects (Pïijđ0.05). We apologize for confusing the reviewer by the figures, and we have re marked them in the manuscript. 21. Line 208: ". . . Total PLFA content in the north-facing slope was 50 and 62% higher than those. . ." – I suppose that 50 and 62% are incorrect – according to the Fig. 2B. R: Total PLFA contents were 20.88 mg kg-1 in the north-facing slope, 13.93 mg kg-1 in the south-facing slope, and 12.89 mg kg-1 in

the northeast-facing slope, respectively. So the total PLFA content in the north-facing slope was 50 ((20.88-13.93)/13.93*100/100) and 62% ((20.88-12.89)/12.89*100/100) higher than those in the south- and northeast-facing slopes, respectively. 22. Line 213: "…G+ PLFA content did not differ significantly among the slope aspects (Fig. 2B)." – according to the Fig. 2B, it is not true. R: The statistical results of G+ PLFA content indicated that there was no significant difference among different slope aspects (Pïïjd0.05). We apologize for confusing the reviewer by the figures, and we have re marked them in the manuscript. 23. Line 216: "… which were 49 and 117% higher…" – I suppose that 49 and 117% are incorrect? According to the Fig. 2B. R: We are very sorry for our negligence. Actinomycete PLFA contents were 1.28 mg kg-1 in the north-facing slope, 1.47 mg kg-1 in the south-facing slope, and 0.59 mg kg-1 in the northeast-facing slope, respectively. Actinomycete PLFA contents had higher contents in the south- and north-facing slopes, which were 149 ((1.47-0.59)/0.59*100/100) and 117% ((1.28-0.59)/0.59*100/100)) higher, respectively, than that in the northeast-facing slope. We have revised it in the manuscript. Line 238 24. Lines 219-221: these two sentences can be summarized. R: As suggested by the reviewer, we have merged these two sentences: "The REs for total, G+, G-, bacterial, actinomycete PLFA contents in the northeast-facing slope were highest among the slope aspects." Line 241-243 25. Lines 173-175: you have measured several characteristics (pH, SAP, WSOC, and so on), but in results section there are no data. Could you, please, include a table or figure with these characteristics? R: Considering the reviewer's suggestion, we have made a table for these characteristics and added it in the manuscript. Table 3. Characteristics of the rhizospheric and non-rhizospheric soils. Slope aspect pH Water Content (100%) SOC (g kg-1) SAP (mg kg-1) NO3 (mg kg-1) NH4 (mg kg-1) WSOC (mg kg-1) WNO3 (mg kg-1) WNH4 (mg kg-1) Rhizospheric soil South-facing 8.55 7.73 9.20 3.23 8.60 12.94 59.12 1.55 0.61 North-facing 8.72 10.37 7.36 2.41 9.70 9.87 37.02 1.27 0.44 Northeast-facing 8.63 10.60 5.21 1.98 7.33 9.05 45.32 1.77 0.53 Non-rhizospheric soil South-facing 8.54 8.13 5.53 1.35 4.93 12.32 38.14 1.20 0.50 North-facing 8.58 10.31 4.90 1.37 6.73 13.13 36.47 0.98 0.38 Northeast-facing 8.58 10.45 4.27 1.68 6.27 12.42

40.39 1.38 0.45 26. Line 275 "...supporting our hypothesis 2..."could you write it more precisely. R: As suggested by the reviewer, we have revised the sentence in the manuscript: "supporting our hypothesis that soil C and N are the main soil nutrient factors that affect RS and NRS microbial communities". Line 296-298 27. Line 278: what do you mean- bioenergetic status - could you explain in your opinion? R: Yes, the "bioenergetic status" here mainly refers to the status of utilization of energy substrates by soil microorganism, in particular, the energy change in the process of using soil organic carbon. 28. Line 291: "These different results may BE due to the differences" R: As suggested by the reviewer, we have revised it in the manuscript: "These different results may be due to the differences...." Line 313 29. There is not so much discussion about north-east slopes in the text- could you add it? R: As the change of soil microbial characteristics at northeast-facing slope were between north-facing and south-facing slopes, and the differences in soil microbial characteristics between north-facing and south-facing slopes were more obvious, so the discussion part paid more attention to north-facing and south-facing slopes. 30. Lines 318-319: this statement is not correct according to table 3. R: We are very grateful to the reviewer's conscientious, and very sorry for our carelessness. We have revised it in the manuscript: "The F/B ratio in our study was highest in the south-facing slope and lowest in the north-facing slope for both RS and NRS". Line 339-341 31. Line 315-324. If soil moisture is an important environmental factor affecting the composition of microbial communities, I suppose you should add your data of soil moisture to manuscript. R: We agree with the reviewer's viewpoint, but we have no monitoring data of soil moisture content in sampling sites. However, the soil moisture contents of sampling sites during our investigation have been added in the Table 3. Table 3. Characteristics of the rhizospheric and non-rhizospheric soils. Slope aspect pH Water Content (100%) SOC (g kg-1) SAP (mg kg-1) NO3 (mg kg-1) NH4 (mg kg-1) WSOC (mg kg-1) WNO3 (mg kg-1) WNH4 (mg kg-1) Rhizospheric soil South-facing 8.55 7.73 9.20 3.23 8.60 12.94 59.12 1.55 0.61 North-facing 8.72 10.37 7.36 2.41 9.70 9.87 37.02 1.27 0.44 Northeast-facing 8.63 10.60 5.21 1.98 7.33 9.05 45.32 1.77 0.53 Non-rhizospheric soil South-facing 8.54

8.13 5.53 1.35 4.93 12.32 38.14 1.20 0.50 North-facing 8.58 10.31 4.90 1.37 6.73 13.13 36.47 0.98 0.38 Northeast-facing 8.58 10.45 4.27 1.68 6.27 12.42 40.39 1.38 0.45 32. Sections 4.2.1 and 4.2.2 -please, analyze data of G+ and G- PLFA contents either in section 4.2.1 or 4.2.2. R: Considering the reviewer's suggestion, we have added an analysis of G+ and G− PLFA contents in section 4.2.1: "Previous studies have found that wetter soils are more enriched in G+ bacteria (Zhang et al., 2005; Drenovsky et al., 2010; Ma et al., 2015), in agreement with the result of the RE G+ PLFA content, but not the NRE G+ PLFA content. As RE was significantly affected by slope aspect for the G+ PLFA content, this may be one of the reasons that caused the difference between RE and NRE G+ PLFA contents. Furthermore, the RDA indicated that the RS WSOC was well correlated with the RE G+ PLFA content, and the NRS WNH4 content was well correlated with the NRE G+ PLFA content (Fig. 4A, B). Although drier soils tend to be more enriched in G− bacteria (Zhang et al., 2005; Drenovsky et al., 2010; Ma et al., 2015), both the higher RE and NRE G− PLFA contents were recorded at the north-facing slope. It has been shown that drier soils can lead to low root exudates, which may lead to lower activities of the soil microorganisms (Zhang et al., 2015)." Line 352-361 33. Line 339-340: "NRS actinomycete PLFA content, however, was lower in the northeast-facing slope than that in the south-facing slope."- but also - than in the north-facing slope, or not? R: Yes, it is. Considering the reviewer's suggestion, we have revised the sentence to be: "NRS actinomycete PLFA content, however, was lower in the northeast-facing slope than those in the north-facing and south-facing slopes." Line 371-372 34. Lines 347-348: "Drier soils tend to be more enriched in G-bacteria and fungi,. . ."? it is not correct according to fig. 2. R: We are very grateful to the reviewer's reminding, and have deleted the sentence. 35. Line 408: the surname SCHAEPMAN⅘ARSTRUB should be written in lower case. R: As suggested by the reviewer, we have revised it in the manuscript: "Schaepman‐strub". 36. Journal names should be abbreviated according to the ISI Journal Title Abbreviations Index (according to Manuscript preparation guidelines for authors). R: We are very grateful to the reviewer's reminding, and the journal names have been abbreviated in the

manuscript. For example, Anderson, T.-H., and Domsch, K.: The metabolic quotient for CO2 (qCO2) as a specific activity parameter to assess the effects of environmental conditions, such as pH, on the microbial biomass of forest soils, Soil Biol. Biochem., 25, 393-395, doi:10.1016/0038-0717(93)90140-7, 1993. 37. Please, add DOI to references. R: As suggested by the reviewer, we have added the DOI to references in the manuscript. For example, Anderson, T.-H., and Domsch, K.: The metabolic quotient for CO2 (qCO2) as a specific activity parameter to assess the effects of environmental conditions, such as pH, on the microbial biomass of forest soils, Soil Biol. Biochem., 25, 393-395, doi:10.1016/0038-0717(93)90140-7, 1993. 38. The quality of Figures 2 and 3 is very poor. Nothing is clear at the picture. Please make columns larger and readable. Could you explain what different letters above the bars (a, b, ab) mean? And when there are no letters? what does it mean? (fig. 1-3). R: Considering the reviewer's suggestion, we have split Figure 2 into 3 figures, and Figure 3 into 4 figures to improve their readability. Different letters above the bars in the figures indicate significant differences at P=0.05. For example, in Figure 1A, NRS MBC content in the north-facing slope was higher than those in the south- and northeast-facing slopes, and there was no significant difference between south-facing slope and northeast-facing slope, so the letter above the column of north-facing slope was a, both the letters above the columns of north- and northeast-facing slopes were b. We are sorry for making the reviewer confused. We have added the same letter (a) above the bars that had no letters before in all figures, and the same letters meant that there was no difference among the slope aspects. 39. Figure 4: RQ (respiratory quotient) ?what is it? Please, make this figure larger and readable. R: The respiratory quotient was BR/MBC. We have changed the "RQ" to "BR/MBC" in Figure 4. To improve its readability, Figure 4 has re plotted and rearranged. Nevertheless, I found this paper of good quality and after correcting it can be publish. R: Once again, we thank the reviewer for the positive comments, and we have revised the manuscript in accordance with reviewer's suggestions.

Bardelli, T., Gómez-Brandón, M., Ascher-Jenull, J., Fornasier, F., Arfaioli, P., Francioli, D., Egli, M., Sartori, G., Insam, H., and Pietramellara, G.: Effects of slope

exposure on soil physico-chemical and microbiological properties along an altitudinal climosequence in the Italian Alps, Sci. Total Environ., 575, 1041-1055, doi:10.1016/j.scitotenv.2016.09.176, 2017. Dearborn, K. D., and Danby, R. K.: Aspect and slope influence plant community composition more than elevation across forest–tundra ecotones in subarctic Canada, Journal of Vegetation Science, 28, 595-604, doi:10.1111/jvs.12521, 2017. Drenovsky, R. E., Steenwerth, K. L., Jackson, L. E., and Scow, K. M.: Land use and climatic factors structure regional patterns in soil microbial communities, Global Ecol. Biogeogr., 19, 27-39, doi:10.1111 / j.1466-8238.2009.00486.x, 2010. Liu, M., Zheng, R., Bai, S., and Wang, J.: Slope aspect influences arbuscular mycorrhizal fungus communities in arid ecosystems of the Daqingshan Mountains, Inner Mongolia, North China, Mycorrhiza, 27, 189-200, doi:10.1007/s00572-016-0739-7 2017. Ma, L. N., Guo, C. Y., Lü, X. T., Yuan, S., and Wang, R. Z.: Soil moisture and land use are major determinants of soil microbial community composition and biomass at a regional scale in northeastern China, Biogeosciences, 12, 2585-2596, doi:10.5194/bg-12-2585-2015, 2015. Zhang, C., Liu, G. B., Xue, S., and Wang, G. L.: Changes in rhizospheric microbial community structure and function during the natural recovery of abandoned cropland on the Loess Plateau, China, Ecol. Eng., 75, 161-171, doi:10.1016/j.ecoleng.2014.11.059, 2015. Zhang, W., Parker, K., Luo, Y., Wan, S., Wallace, L., and Hu, S.: Soil microbial responses to experimental warming and clipping in a tallgrass prairie, Global Change Biol., 11, 266-277, doi:10.1111/j.1365-2486.2005.00902.x, 2005.

Please also note the supplement to this comment:
https://www.solid-earth-discuss.net/se-2017-137/se-2017-137-AC2-supplement.pdf
* * *
SED

Interactive
comment

Feb 17, 2018

Solid Earth

Dear Reviewer,

This letter certifies that I have edited the language of the manuscript entitled "Influence of slope aspect on the microbial properties of rhizospheric and non-rhizospheric soil on the Loess Plateau, China". The language of the manuscript should meet your standards.

Sincerely,

Dr. William Blackhall

Editor, Global Biological Editing

editor@globalbiologicalediting.com

www.globalbiologicalediting.com

**Fig. 1.**
1   **Table 3.** Characteristics of the rhizospheric and non-rhizospheric soils.

| | Slope aspect | pH | Water Content (100%) | SOC (g kg$^{-1}$) | SAP (mg kg$^{-1}$) | NO$_3$ (mg kg$^{-1}$) | NH$_4$ (mg kg$^{-1}$) | WSOC (mg kg$^{-1}$) | WNO$_2$ (mg kg$^{-1}$) | WNH$_4$ (mg kg$^{-1}$) |
|---|---|---|---|---|---|---|---|---|---|---|
| Rhizospheric soil | South-facing | 8.55 | 7.73 | 9.20 | 3.23 | 8.60 | 12.94 | 59.12 | 1.55 | 0.61 |
| | North-facing | 8.72 | 10.37 | 7.36 | 2.41 | 9.70 | 9.87 | 37.02 | 1.27 | 0.44 |
| | Northeast-facing | 8.63 | 10.60 | 5.21 | 1.98 | 7.33 | 9.05 | 45.32 | 1.77 | 0.53 |
| Non-rhizospheric soil | South-facing | 8.54 | 8.13 | 5.53 | 1.35 | 4.93 | 12.32 | 38.14 | 1.20 | 0.50 |
| | North-facing | 8.58 | 10.31 | 4.90 | 1.37 | 6.73 | 13.13 | 36.47 | 0.98 | 0.38 |
| | Northeast-facing | 8.58 | 10.45 | 4.27 | 1.68 | 6.27 | 12.42 | 40.39 | 1.38 | 0.45 |

**Fig. 2.**

[Figure]

**Fig. 2** The rhizospheric effects of MBC, BR, SIR, and PLFA contents. Error bars are standard errors (n=3). Different letters above the bars indicate significant differences at $P$=0.05.

**Fig. 3.**

---

## Author Response (AR2)

Dear Editors and Reviewers,

We are very glad to receive your email with regard to our manuscript **se-2017-137**, entitled "**Influence of slope aspect on the microbial properties of rhizospheric and non-rhizospheric soil on the Loess Plateau, China**". The comments from the reviewers are very helpful for revising and improving our manuscript, as well as hold great guiding significance to our researches. We take all of these comments into account in preparing the revised manuscript. We believe that the manuscript has been improved satisfactorily and hope it will be accepted for publication in **Solid Earth**.

We thank the editors and reviewers again for all the work you have done for our manuscript. If you require any further information, please contact with us at any times. All the changes in the manuscript have been listed below and marked in the manuscript.

**Report #1**

General comments

The manuscript has largely improved from the original (discussion) version and I therefore recommend publication after minor revision. See below some minor comments.

R: We are very grateful to the reviewer for his recognition of our work and helpful comments.

Section 2.2 Could you be more specific on the selection of these aspects and not including others?

R: Considering the reviewer's suggestion, we have added a sentence to explain e the representativeness of these slope aspects: "which had the same site conditions (all with *Artemisia sacrorum* as the dominant species, same rehabilitation age, geographical proximity, etc.) and represented sunny slope, half-sunny slope and shady slope, respectively." Lines 150-152

Table 3. Please clarify what all the abbreviations of the soil variables mean. Also please specify whether these are average values. Did you measure standard deviation? How many replicates (n=3)?

R: As suggested by the reviewer, we have already added the relevant information in the Table 3: "SOC: soil organic carbon, SAP: available phosphorus, $NO_3$: nitrate nitrogen, $NH_4$: ammonium nitrogen, WSOC: water-soluble organic carbon, $WNO_3$: water-soluble nitrate nitrogen, $WNH_4$: water-soluble ammonium nitrogen. The above data are average values (n=3)". We did not measure the standard deviation here. Lines 204-205

**All relevant changes**

1. A new affiliation had been listed in the second position of the list of the affiliations for the first author, after the consent of all the authors of this manuscript. Lines 4-14

2. We are very sorry for our negligence: the "northeast" should be the "northwest". And we have changed them in the manuscript, including the Figures 1-3.

3. We have updated the funds in the Acknowledgements section. Lines 418-420

[revised manuscript text omitted]

---

## Author Response (AR3)

Dear Editors,

We are very glad to hear from you that our manuscript (**se-2017-137**) has been accepted for publication in **Solid Earth**. We are very satisfied with the review process, and thank the editors for all the work you have done for our manuscript.

Non-public comments to the Author: In the added sentence, remove "etc".

R: As suggested by the editor, we have deleted the "etc" in the manuscript.